# Unravelling genetic differentiation between *Glossina brevipalpis* populations from two distant National Parks in Mozambique

Denise R. A. Brito[1,2]*, Adeline Ségard[3], Fernando C. Mulandane[1], Nióbio V. Cossa[1,2], Hermógenes N. Mucache[1,4], Sophie Ravel[3], Thierry De Meeûs[3�]*, Luis Neves[1,2�]

1 Department of Genetic Characterization of Populations and Biodiversity, Biotechnology Centre, Eduardo Mondlane University, Maputo, Mozambique, 2 Department of Veterinary Tropical Diseases, University of Pretoria, Pretoria, South Africa, 3 Intertryp, Univ Montpellier, Cirad, IRD, Montpellier, France, 4 Veterinary Faculty, Eduardo Mondlane University, Maputo, Mozambique

උ Contributed equally.
* thierry.demeeus@ird.fr (TDM); denise.brito@uem.mz (DRAB)

## Abstract

African trypanosomosis (AT), caused by protozoan parasites of the genus *Trypanosoma*, has plagued the African continent for centuries, affecting both humans and animals. Its principal vector, tsetse flies, can be found across sub-Saharan Africa. Vector control represents an efficient way to reduce the burden of AT. In Mozambique, control campaigns reshaped tsetse fly distribution to what it is today, with four species presently found: *Glossina brevipalpis, G. pallidipes, G. morsitans* and, *G. austeni*. Additionally, *G. brevipalpis* can be observed in two National parks, Gorongosa National Park in the Centre and Maputo National Park in the South, with an 840 km wide tsetse-free zone between them. In order to improve our knowledge on the genetic diversity in these populations, and their probable isolation, we undertook a population genetics study with 11 microsatellite loci. We found that these two zones behave as strongly isolated subpopulations, only exchanging a few individuals per year. To explain this finding, we suggest the existence of undocumented pocket populations between the two parks, or, in the absence of these, the accidental translocation of tsetse flies during human-driven animal transportation. We suggest that translocation through human-driven animal movement should be explored in future studies investigating *Glossina* populations. If eradication were to be attempted, re-invasion of the tsetse via motorized human transport should be considered in conjunction with the exploration of other sites within a 30 km radius to validate that no sources of re-invasion exist around these parks.

## Author summary

African trypanosomoses (AT), is caused by single celled parasites of the genus *Trypanosoma*. These have plagued the African continent for centuries, affecting

**Data availability statement:** All relevant data are within the manuscript and its Supporting Information files.

**Funding:** This project has received funding from the European Union's Horizon 2020 research and innovation program under grant agreement n°101000467, acronym ''COMBAT'' (Controlling and Progressively Minimizing the Burden of Animal Trypanosomosis). The funders had no role in study design, data collection and analysis, decision to publish, or preparation of the manuscript.

**Competing interests:** The authors have declared that no competing interests exist.

both humans and animals. Its principal vector, tsetse flies, can be found across sub-Saharan Africa. Vector control represents an efficient way to reduce the burden of AT. In Mozambique, control campaigns reshaped tsetse fly distribution to what it is today, with four species presently found: *Glossina brevipalpis, G. pallidipes, G. morsitans* and, *G. austeni*. Additionally, *G. brevipalpis* can be observed in two National parks, Gorongosa National Park in the Centre and Maputo National Park in the South, with an 840 km wide tsetse-free zone between them. In the present work, we have analysed the genetic variation of this tsetse fly within and across these two parks. We found that these two zones behave as strongly isolated subpopulations, only exchanging a few individuals per year. Within parks, average dispersal appeared to reach 30 km per generation. To explain this finding, we suggest the existence of undocumented pocket populations between the two parks. In the absence of such pockets, the most probable explanation is the accidental translocation of tsetse flies during human-driven animal transportation. If eradication were to be attempted, re-invasion of the tsetse via motorized human transport should be considered in conjunction with the exploration of other sites within a 30 km radius to validate that no sources of re-invasion exist around these parks.

## Introduction

African trypanosomosis (AT), caused by protozoan parasites of the genus *Trypanosoma*, has plagued the African continent for centuries, affecting both humans and animals, causing a heavy economic burden [1–4]. These blood parasites have a diverse range of mammalian hosts [1]. Their primary biological vectors are the hematophagous flies known as tsetse (*Glossina* spp). Control of tsetse flies has been conducted over the last and present century so as to reduce the burden caused by both human (HAT) and animal (AAT) African trypanosomosis [3,5,6].

 *Glossina* species are classified into three main groups (*Fusca*, *Palpalis* and *Morsitans*). These presently include a total of 31 species/subspecies adapted to specific habitats [3,5,7]. The *Fusca* group corresponds to forest-dwelling flies that are mostly found in western-central Africa, except for *Glossina longipennis* and *G. brevipalpis,* which are located in eastern and southern Africa [3,5]. Considering the case of Mozambique, four species of tsetse flies are found in the country, namely *G. austeni*, *G. morsitans* and *G. pallidipes* from the Morsitans group and *G. brevipalpis* of the Fusca group [8–10]. In this country, the southern fly belt consists of two species (*G. austeni* and *G. brevipalpis*), while the central and northern belts are composed of the four tsetse species [8–10].

 In the late 1890s, tsetse fly populations suffered a significant reduction, partially due to the occurrence of a rinderpest pandemic that drastically diminished the populations of both wild and domestic ungulates, particularly in southern Africa, including Mozambique [5,11]. Tsetse flies survived in small pockets, allowing for a subsequent reoccurrence and expansion of the extremely reduced *Glossina* populations once

the presence of game and cattle was re-established [5,11]. Nevertheless, in South Africa and Mozambique, this event was followed, between 1920 and 1970, by important vector control programs against all tsetse fly species [12–14]. These control campaigns reshaped the tsetse distribution to what it is today, with the dominance of *G. brevipalpis* and *G. austeni* in several zones of Southern Africa, and in particular in Mozambique [11–13,15–17].

*Glossina brevipalpis* has been detected in Kenya, Rwanda, Tanzania, Zambia, Mozambique and South Africa [18]. The distribution of *G. brevipalpis* is limited by its habitat (forests with bush clumps for its breeding sites) and climate. It is specifically influenced by temperature (16 °C to 32 °C) and relative humidity (≥ 70% of relative humidity) [3,14,15]. In Mozambique, the southern provinces, Gaza and Inhambane, have remained tsetse-free for the last 120 years, without evidence of tsetse fly reinvasion. This separates the central population of *G. brevipalpis*, mainly in the Gorongosa National Park, from the southern population of the same species, primarily situated in the Maputo National Park [18] (Fig 1).

One very efficient strategy to reduce trypanosomosis incidence is vector control [19]. In order to design the most efficient control strategy, understanding the dynamics of the existing populations of *Glossina* is important [20]. It is not only essential to know the existing species and their distribution within a country, but also to understand the genetic structure and gene flow between their populations [20,21]. The use of polymorphic genetic markers and population genetic tools can provide key information on the structure of targeted populations, reproductive strategies, gene flow, population sizes and dispersal distances. It can thus lead to designing the best strategies for control [20,21]. For this purpose, microsatellites have been extensively used for tsetse fly population genetics over the last two decades [21–25].

The present paper aims to assess the genetic diversity and gene flow within and between the central and southern Mozambique populations of *G. brevipalpis* located in the Gorongosa and Maputo National parks respectively.

## Methods

### Ethics statement

All applicable international, national, and/or institutional guidelines for the care and use of animals were followed and all procedures performed in studies involving animals were in accordance with the ethical standards of Biotechnology Centre – Eduardo Mondlane University and the practice at which the study was conducted (Animal Research Ethics Committee of the Biotechnology Centre of the Eduardo Mondlane University (CEPA-CBUEM); approval number: CEPA-CBUEM 05/2023). The sampling procedures reported herein were authorized by the respective park authorities: For Gorongosa National Park the Department of Scientific Services of Gorongosa National Park issued the permit PNG/DSCi/C231/2022; For Maputo National Park: permit 03/01/2022 issued by the National Administration for the Conservation Areas.

### Study sites and sample collection

Gorongosa National Park (GNP) is located in the central region of Mozambique. It was originally established as a hunting reserve in the 1920s and now includes 3,770 km$^2$ of protected land, which is composed of a mix of forest and savannah landscapes, with a high diversity of fauna and flora. This area includes four species of tsetse flies, namely *G. pallidipes, G. morsitans, G. brevipalpis* and *G. austeni*. About 840 km to the south, in the most southern point of the country, there is a tsetse hot spot that partially overlaps with the Maputo National Park (MNP). The park was established in 1960 as a national reserve to protect the extensive population of elephants found in the region. Today, it covers a total of 1,718 km$^2$, comprising various ecosystems including forests that harbour two species of tsetse flies, *G. brevipalpis* and *G. austeni*. Between these two parks, including northern Maputo province, Gaza and Inhambane provinces, a putatively tsetse-free area has existed since the rinderpest pandemic of 1896 [12]. For this reason, these two parks were chosen as sampling areas to study gene flow of *G. brevipalpis* populations between the south and central regions of Mozambique.

Six hundred and seventy-nine (679) *G. brevipalpis* specimens were collected in GNP (342) and MNP (337) using 49 traps (36 H-traps [26] and 8 NGUs traps [27], enhanced with odour attractants (1-octen-3-ol and acetone) [26–28]). We

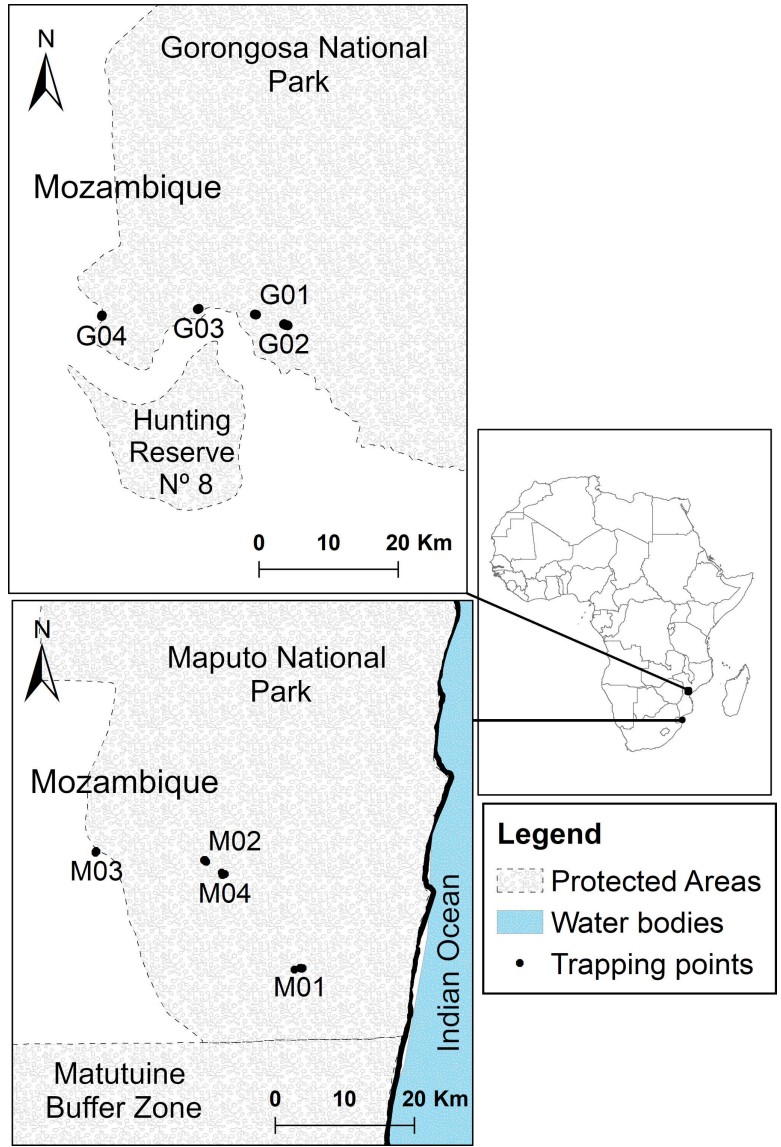

**Fig 1. Map of sampling sites (indicated by black dots) of *Glossina brevipalpis* in both Gorongosa National Park (G01–G04) and Maputo National Park (M01–M04).** Traps contained in each site and their precise GPS coordinates can be seen in S1 Table. (This map was produced from base map imported from The Humanitarian Data Exchange - https://data.humdata.org/dataset/cod-ab-moz).

deployed H traps in the two parks to capture *G. brevipalpis*. However, we also deployed NGU traps to capture other *Glossina* species for another study, which caught some *G. brevipalpis* individuals used in the present study. All flies were captured between June 14 and July 22 2022 corresponding to less than one tsetse fly generation [7]. Twenty-nine (29) traps were deployed in GNP and 20 in MNP (Fig 1). In each park, we selected four different sites in which two to six traps were placed. Neighbouring traps covered an average distance of 272 m with a minimal distance of 150 m and a maximum of 516 m (Fig 1). All traps were georeferenced using the eTrex 10 GPS (Garmin Ltd., USA) (See S1 Table). The overall sampling transects, i.e., the distance between the most extreme traps, reached 25 km in GNP and 14 km in MNP. Tsetse flies were identified using morphological criteria [29].

## DNA extraction and genotyping

Four hundred and two (402) *G. brevipalpis* (GNP – 202 and MNP – 200) were selected for genotyping based on the location and condition of specimens (S1 Table). DNA was extracted from 3 legs per individual using the chelex method [30] after which the DNA was diluted for amplification. We used 11 microsatellite primers developed by Gstöttenmayer et al., (2023) [31] to genotype individual flies collected from both parks: Gb5, Gb28, Gb35, Gb48, Gb66, Gb70, Gb72, Gb73, Gb92, Gb158, and Gb165. To incorporate the dyes into the PCR product, microsatellite forward primers contained the M13 adapter sequence, and amplification was done with M13 primers attached to one of these four dyes: VIC, NED, PET and FAM. PCRs were carried out in a 20 µL reaction with 1x PCR Buffer, 0.2 mM DNTPs, 0.08 µM forward primer, 0.1 µM reverse primer, 0.1 µM M13 primer with dye, 0.5 U Taq polymerase and 10 µL of diluted DNA. The PCR cycling conditions were as follows: 95 °C for 3 mins and then 10 cycles of 95 °C for 30 s, $T_a$ (58/59 °C) + 5 °C for 30 s, 72 °C for 1 min dropping 0.5 °C each cycle, then 30 cycles of 95 °C for 30 s, $T_a$ (58/59 °C) for 30 s, 72 °C for 1 min, with final elongation step at 72 °C for 5 min. The PCR products were pooled by four (the different dyes) and resolved on an ABI 3500XL sequencer (Thermo Fisher Scientific, USA). Allele calling was done using GeneMapper Software v6.1 [32] and the size standard GS600LIZ (Thermo Fisher Scientific, USA).

## Data analysis

Genotypic data were formatted into the appropriate file types using Create v1.37 [33]. Analyses were done using 10 autosomal loci developed by Gstöttenmayer et al., (2023) [31] (S2 Table). As Gb70 is sex-linked [31], it was removed from our dataset.

## Quality testing of the sampling

To describe the population genetic structure of *G. brevipalpis*, we used Wright's *F*-statistics [34]: $F_{IS}$, which measures the effect of deviation from random mating within subpopulations on inbreeding, and $F_{ST}$, which measures the effect of subdivision. These were respectively estimated with Weir and Cockerham's (1984) [35] unbiased estimators (θ and *f*). Their significant deviation from 0 was tested with permutations of alleles between individuals within subsamples (for $F_{IS}$) and individuals between subsamples (for subdivision). In each case, the statistic used was the $F_{IS}$ estimator or the natural logarithm of the maximum likelihood ratio (*G*) [36], respectively. We also tested linkage disequilibrium between each pair of loci with the *G*-based randomization test over all subsamples [37]. For each test, we implemented 10,000 permutations. Estimates and testing were undertaken with Fstat v2.9.4 [38].

A Wahlund effect occurs when individuals belonging to subpopulations with different allele frequencies are admixed into the same subsample. This was first used to explain heterozygote deficits as compared to Hardy-Weinberg expectations following this phenomenon [39]. Nonetheless, Wahlund effects can also minimize subdivision measures, and increase linkage disequilibrium between loci. To determine the levels at which *G. brevipalpis* from Mozambique were subdivided, we used the Wahlund effect detection technique of Goudet et al., (1994) [40]. For this, we considered four sampling designs, depending on the level considered: trap, location, park, and all flies together (All). If a Wahlund effect occurs at one level, a significant change should be observed. We thus computed $F_{IS\text{-trap}}$, $F_{IS\text{-location}}$, $F_{IS\text{-park}}$, and $F_{IS\text{-All}}$, and compared them using the Wilcoxon signed rank test [41] for paired data using R-commander package v2.9-5 (Rcmdr) [42,43] for R v4.4.2 [44], with the alternative hypotheses: $F_{IS\text{-trap}} < F_{IS\text{-location}} < F_{IS\text{-park}} < F_{IS\text{-All}}$. We also compared $F_{ST}$'s measured at these levels (except All). For those, we undertook the same test but with a reverse alternative hypothesis. We finally compared the proportions of locus pairs in significant LD with one-sided Fisher exact tests [45] with R v4.4.2 (command fisher.test). Because of the non-independent test series undertaken for each parameter compared, we needed to adjust the obtained *p*-values with the Benjamini and Yekutieli (BY) procedure [46], with the command "p.adjust" in R v4.4.2.

## Quality testing of the loci

We first checked the statistical independence between each locus pair with the $G$-based randomization test at the BY level of significance. We also examined the deviations from expected genotypic frequencies with $F_{IS}$ estimates and testing as described above. We additionally computed 95% confidence intervals with 5,000 bootstraps over loci for the average of $F_{IS}$ and $F_{ST}$ with Fstat v2.9.4. To obtain 95% CI of $F_{IS}$ for each locus, we used 5,000 bootstraps of individuals in each subsample for each locus with Genetix v4.05.2 [47]. We computed the average of $F_{IS}$ and its 95% CI for each locus obtained with Genetix v4.05.2 weighted by the number of visible genotypes and the local genetic diversity as estimated by Nei's unbiased estimator ($H_s$) estimated with Fstat v2.9.4. We estimated the variation of $F_{IS}$ and $F_{ST}$ across loci with the standard error (SE($F_{IS}$) and SE($F_{ST}$)) obtained by jackknives over loci with Fstat v2.9.4. The presence of null alleles was assessed with several criteria as described elsewhere [48,49]. In the case of null alleles, the ratio $r_{SE} = \text{SE}(F_{IS})/\text{SE}(F_{ST})$, a positive correlation is expected between $F_{IS}$ and $F_{ST}$, and between the number of missing data ($N_{missing}$) and $F_{IS}$. Correlations were assessed and their significance tested with one-sided Spearman's rank correlation using Rcmdr v2.9-5. Null allele frequencies ($p_{nulls}$) were then estimated with the EM algorithm [50] using FreeNA [51]. For this, we recoded missing data as homozygotes for allele 999 as advised [51], only for loci for which missing data indeed corresponded to true null homozygotes (see Results section). Finally, we undertook the regression $F_{IS} \sim p_{nulls}$ with its 95% CI of bootstraps and computed its determination coefficient and the intercept, which should correspond to the value of $F_{IS}$ in the absence of null alleles ($F_{IS-0}$).

## Analyses of population subdivision and gene flow

We estimated subdivision with FreeNA and the ENA algorithm to correct for null alleles ($F_{ST-ENA}$). For this, we recoded missing genotypes suspected to correspond to true null homozygotes as 999999 as recommended [51]. Microsatellite loci generally display a high level of polymorphism, leading $F_{ST}$ to reflect mutation and immigration together. To correct for this excess of polymorphism, we used the $F_{ST}'$ approach [52,53]. The maximum possible $F_{ST}$ ($F_{ST-max}$) was computed with Fstat v2.9.4 after allele recoding by RecodeData v0.1 [53], which was used to compute $F_{ST-ENA'} = F_{ST-ENA}/F_{ST-max}$. Meirmans and Hedrick proposed an alternative correction ($G_{ST}''$) [54] which is expected to be more accurate. Nevertheless, $G_{ST}''$ neither allows correction for null alleles nor the estimate of 95% CIs with available programs.

We estimated the probable number of immigrant flies exchanged between the two parks as $N_em = (1-F_{ST-ENA'})/(4F_{ST-ENA'})$, assuming an infinite Island model ($n$=infinite), or $N_em = (1-F_{ST-ENA'})/(8F_{ST-ENA'})$, assuming a two-island model ($n$=2) [55].

## Effective population sizes

We estimated effective population sizes with five methods: the heterozygote excess method ($H_{ex}$) from De Meeûs and Noûs, (2023) [56]; the linkage disequilibrium method (LD) [57] corrected for missing data [58]; the Coancestry method (CoA) [59]; the one locus and two loci correlation method (1L2L) [60]; and the sibship frequency method (Sib) [61]. For $H_{ex}$ we estimated $N_e$ from the published formula in a spreadsheet program from $F_{IS}$ values obtained in each subsample for each locus and averaged across loci as recommended [56]. For LD and CoA, we used NeEstimator v2.1 [62]; for 1L2L we used Estim v1.2 [60]; and finally, for Sib, we used Colony [63]. We also kept the minimum (min) and maximum (max) values and averaged all estimates across methods by weighting those with the number of usable figures obtained, as recommended [64]. In case of unusable values (i.e., "Infinite"), and in order to compare effective population sizes between the two parks, we retrieved the 95% CI outputted by the softwares used (bootstraps over individuals for $H_{ex}$, 1L2L, and Sib, jackknives for CoA and parametric for LD). As suggested by Waples [65], we then replaced all "Infinite" by a very big figure (here we chose 10 times the maximum observed one, 30000), and computed harmonic means of all $N_e$ and 95%CI across methods and overall.

## Time of isolation or gene flow between the two parks

We used the averaged effective population size estimate to compute the number of generations of the split between GNP and MNP subpopulations, assuming total isolation, with the formula $t = -2 \times N_e \times ln(1 - F_{ST\text{-}ENA'})$ (e.g., Hedrick (2005), equation 9.13a, p 502 [52]). We also extracted the immigration rate ($m$) from $N_e m$ estimated with $n = 2$ or $n =$ infinite and used it to compute the average dispersal distances covered by tsetse flies per generation, assuming rare transportation between the two parks, as $\delta = m \times D\text{geo}$ and its 95% CI.

## Isolation by distance, sex biased dispersal between traps and bottleneck within each park

We undertook isolation by distance analyses within each park separately (to avoid a two-points regression), and to confirm the free circulation of tsetse flies in these two areas. We used Rousset's model [55] $F_R = a + b \times ln(Dgeo)$ in two dimensions, where $F_R$ is Rousset's genetic distance between two traps, $a$ is the intercept, $b$ is the slope of the corresponding regression and $ln(D_{geo})$ is the natural logarithm of the geographic distance between the two traps. Rousset showed that, in case of isolation by distance $b$ is the reverse of the neighbourhood (number of connected individuals, $Nb = 1/b$). To compute $F_R$, we used the ENA algorithm [51] to correct for the effect of null alleles [66]. Between each pair of traps we computed $F_{ST\text{-}ENA}$ and its 95%CI with 5000 bootstraps over loci with FreeNA [51], and computed geographic distances with the package geosphere [67] for R (command distGeo). We then computed $F_R = F_{ST\text{-}ENA}/(1 - F_{ST\text{-}ENA})$ and its 95%CI and regressed those against $ln(D_{geo})$ to obtain the slope of isolation by distance and its 95%CI.

A genetic signature of sex biased dispersal can only be detected for very big differences between genders, e.g., when one sex is highly philopatric and the other disperse a lot [68]. This thus applies to weakly subdivided populations. We randomized the gender of each fly 10,000 times to test for the significant difference of the corrected assignment index, AIc, its variance, vAIc, and $F_{ST}$. We did these computations and corresponding tests (two-sided) with HierFstat [69] for R v4.4.2.

In order to help the interpretation of results, we also tried to detect a genetic signature of a past bottleneck that would correspond to the rinderpest pandemic that occurred in the years 1898 – 1901, or the massive vector control that was undertaken in the years 1948 – 1970. We implemented this test with the software Bottleneck [70]. As recommended (De Meeûs et al., (2021), p108-109 [71]), a significant bottleneck signature was recognised when the Wilcoxon test outputted significant $p$-values for the infinite allele model (IAM) and the two-phase mutation model (TPM), at least. We obtained a global $p$-value across the two parks with the generalized binomial procedure [72], using $k' = 2$ as recommended [73], with the software MultiTest v1.2 [37].

## Bayesian clustering and neighbour-joining trees

Additionally, we used the Bayesian clustering method implemented by the software STRUCTURE v2.3.4 [74], with a 5,000 burning period and 50,000 MCMC iterations, a number of clusters from 1 to 4 and 10 replicates. We then looked for the optimal partition with the method proposed by Evanno et al.,'s (2005) [75] with the online facility StructureSelector [76]. We hoped that this approach would help elucidate the pattern of dispersal between the two parks through the detection of probable immigrants or individuals that probably descended from immigrant parents a few generations ago.

To validate the result produced by STRUCTURE v2.3.4, we used a Neighbour-Joining (NJ) approach. We computed Cavalli-Sforza and Edward's chord distance [77] corrected for null alleles [51] between all individuals of *G. brevipalpis* from GNP and MNP. The Genetic distance matrix was then used to build an NJTree with MEGA X [78]. Another two trees were built using only individuals with complete genotypes (no missing data). With this alternative dataset, we also used PopTree [79], which uses Nei's $D_A$ genetic distance [80], with 1000 bootstraps. All trees were edited in Interactive Tree of Life (iTOL) v6 [81].

## Results

Of the 402 tsetse flies submitted to genotyping, 396 were successfully genotyped (i.e., with more than 4 loci amplified). These 396 genotypes (196 GNP and 200 MNP) were used for further analyses and are presented in S2 Table.

## Quality testing of the sampling

Significance only occurred for comparisons with $F_{IS}$ and LD, and only between "All" (all traps of the two parks pooled into a single subsample) and each of the three other sampling designs (all $p_{BY} < 0.005$). This means that migration of *G. brevipalpis* is free within the sampling areas in each park, i.e., across 25 km in GNP and 14 km in MNP. Consequently, for subsequent analyses, the level of subpopulation used was the park (i.e., Gorongosa and Maputo).

## Quality testing of the loci

We found five locus pairs in significant LD, one of which (Gb5 and Gb66) remained significant after BY correction ($p_{BY}$ = 0.02). There was a significant heterozygote deficit: $F_{IS}$ = 0.166 in 95% CI = [0.084, 0.256] (*p*-value < 0.0002). It seemed that null alleles explained these figures well. Indeed, $r_{SE} \approx 3$, the correlation between $F_{IS}$ and $F_{ST}$ was positive and significant (0.6848, *p*-value = 0.0175). Nevertheless, the correlation between $N_{missing}$ and $F_{IS}$ was negative ($\rho$ = -0.0788, *p*-value = 0.5943) because of an excess of missing genotypes at seven loci. Indeed, it can be seen from Fig 2, that the correspondence of missing genotypes with homozygotes for null alleles may be true for only three loci (Gb35, Gb73, and Gb165). Consequently, for null allele frequency estimates, we recoded missing data as 999999 only for these three loci. The resulting regression $F_{IS} \sim p_{nulls}$ provided a coefficient of determination ($R^2$) below 0.9 ($R^2$ = 0.8584, $F_{IS-0}$ = 0.0316). This came from two outlier loci displaying too small $F_{IS}$ when compared to the corresponding null allele frequency: loci Gb73 and Gb165. We thus chose to recode missing data for these two loci as "000000" because most blanks probably corresponded to other amplification failures than null alleles (hence only Gb35 remained 999999 for missing genotypes). With this new dataset, we obtained a very good adjustment (Fig 3). The model indeed explained almost 100% of $F_{IS}$ variation across loci, with a negative intercept ($F_{IS-0}$ = - 0.0043 in 95% CI = [- 0.0704, 0.059]), compatible with pangamic populations of average effective population size $N_e$ = 116 in 95% Confidence Interval (CI) = [7, Infinite] [56].

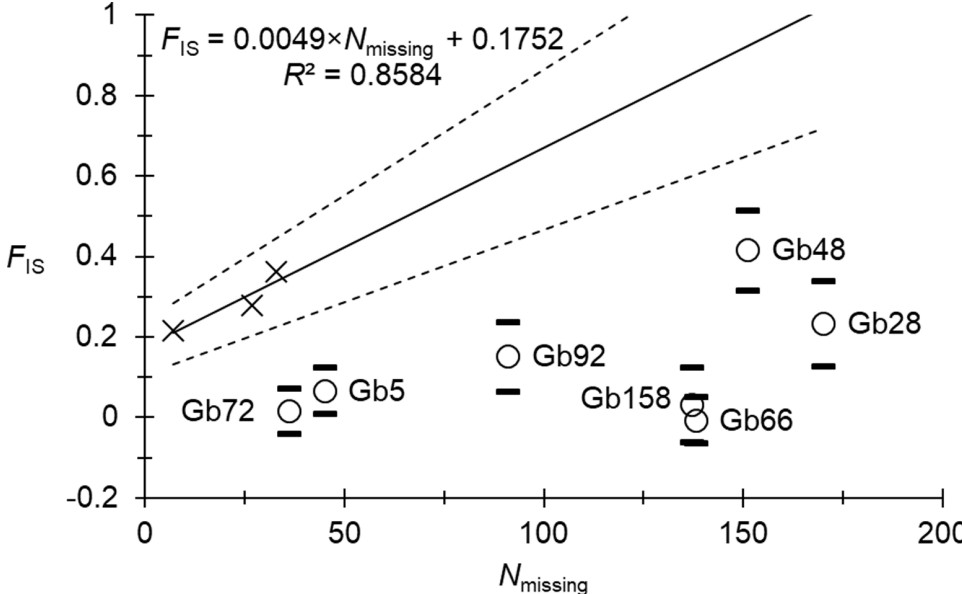

**Fig 2. Regression $F_{IS} \sim N_{missing}$ for *Glossina brevipalpis* from Mozambique.** The average (straight line) and its 95% CI (dotted lines) are represented. Loci for which missing data probably corresponded to null homozygotes only were used for the regression and are indicated with black crosses. Other loci are indicated by their names and empty circles with their 95% CI (black dashes).

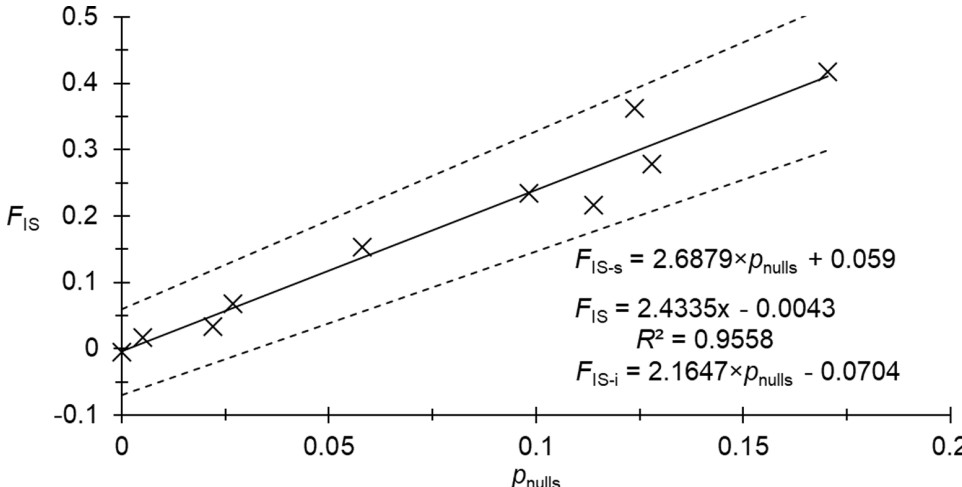

**Fig 3. Regression $F_{IS} \sim p_{nulls}$ for *Glossina brevipalpis* from Mozambique obtained with 10 microsatellite loci with all missing genotypes coded as such except for locus Gb35, for which missing genotypes were considered as null homozygotes (999999) for FreeNA analyses.** The average equation (straight line) and its determining coefficient ($R2$) and of the 95% CI (dotted lines) are represented.

To conclude, two loci, Gb5 and Gb66, are in statistical linkage and thus may introduce redundancy in multi-locus-based tests and parameter estimates. Locus Gb5 displayed a significant heterozygote deficit $F_{IS}=0.0674$ in 95% CI = [0.0082, 0.1244], while locus Gb66 displayed a non-significant excess: $F_{IS}=-0.0058$ in 95% CI = [-0.0636, 0.0506]. Subdivision at these two loci (0.032 and 0.016 respectively) appeared significantly smaller than the average $F_{ST}=0.066$ in 95%CI = [0.038, 0.099]. A glance at https://genome.ucsc.edu/ for *G. brevipalpis* draft genome, and a BLAT [82] of primer sequences of these two loci revealed that these two loci were not found in the same scaffold. It is thus highly unlikely that these sequences are close to each other. The statistical linkage we found may, therefore, be a consequence of selective events driving these two markers correlatively. We thus chose to exclude loci Gb5 and Gb66 from further analyses and kept coding missing genotypes as null homozygotes (999999) for Gb35 only as regard to subdivision estimates with FreeNA.

With the eight remaining loci (Gb28, Gb35, Gb48, Gb72, Gb73, Gb92, Gb158, and Gb165), no locus pair remained significant LD after BY correction (all $p_{BY}$ = 1). There was a significant heterozygote deficit $F_{IS}=0.205$ in 95% CI = [0.114, 0.299] (two-sided $p$-value<0.0002). Additionally, we observed a higher $F_{IS}=0.276$ in MNP than in GNP ($F_{IS}=0.132$) ($p$-value=0.0039, Wilcoxon signed rank test). This probably came from a difference in null allele frequencies, which appeared to be much higher in MNP than in GNP (0.1164 and 0.0531 respectively, $p$-value=0.0039, Wilcoxon signed rank test). Alternatively, genetic diversity did not vary significantly between MNP and GNP ($H_S=0.7398$ and 0.7868, respectively, $p$-value=0.9512).

## Population subdivision and gene flow

We observed a highly significant subdivision between the two parks ($p$-value<0.0001), with $F_{ST-ENA}'=0.2693$ in 95% CI = [0.1631, 0.3964]. With these figures, we could estimate the number of immigrants exchanged between the two subpopulations $N_e m=0.70$ in 95% CI = [1.36, 0.39] individuals per generation with $n$=infinite, and $N_e m=0.35$ in 95% CI = [0.20, 0.68] with $n$=2.

## Effective population sizes

For the sake of consistency in $H_{ex}$ results, and to get less "Infinite" results with that method, we used the averaged $F_{IS}$ from Genetix v 4.05 computations and its 95%CI for each locus and each park. We observed two "infinite" estimates of effective population sizes in MNP (for heterozygote excess and CoAncesty methods), which could not be taken into

account for the average. Consequently, the following estimates probably corresponded to underestimations of the real average effective population size, especially so for MNP. The average effective population size was $N_e = 844$ in minmax = [676, 1012]. GNP displayed a smaller $N_e = 629$ in minmax = [21, 2068] compared to MNP with $N_e = 1202$ in minmax = [110, 3328], but the significance of this difference could not be tested. Nevertheless, for the three methods for which MNP was not "infinite", it always gave a higher $N_e$ than the one in GNP. As can be seen from the Fig 4, harmonic means confirmed this trend, though 95%CI largely overlapped. The arithmetic average appeared significantly higher than what was computed by Gstöttenmayer et al., (2023) [31]. The intercept of the regression of $F_{IS} \sim p_{nulls}$ and harmonic means, also provided significantly smaller figures (Fig 4). This may mean that the real $N_e$ lies between 200 and 1000 individuals (Fig 4).

**Time of isolation or gene flow between the two parks**

With the isolation hypothesis, and with these $N_e$ values, we computed that the subpopulations from the two parks needed to have split 86 years ago with a 95% CI = [47, 138], and with a minmax = [38, 166]. In terms of dates, this would lead to years 1936 in 95% CI = [1884, 1975], and the oldest date going back to 1856. In case of still ongoing exchanges of immigrants, we computed that this would have required an average dispersal distance of less than 1 km per generation for $n = 2$, and between 1 and 2 km per generation for $n =$ infinite.

**Isolation by distance, sex biased dispersal between traps and bottleneck within each park**

In GNP $b = -0.0013$ in 95%CI = [-0.0061, 0.001], and in MNP $b = -0.0009$ in 95%CI = [-0.0010, -0.001], hence a total absence of isolation by distance.

We could not detect any significant genetic signature of sex-biased dispersal. Parameters provided values in different directions and the smallest $p$-value = 0.0566.

There was no obvious signature of a bottleneck, as $p$-value = 0.0365 for the IAM model and $p$-value > 0.25 (maximum possible with $k = 2$) for the TPM and SMM models.

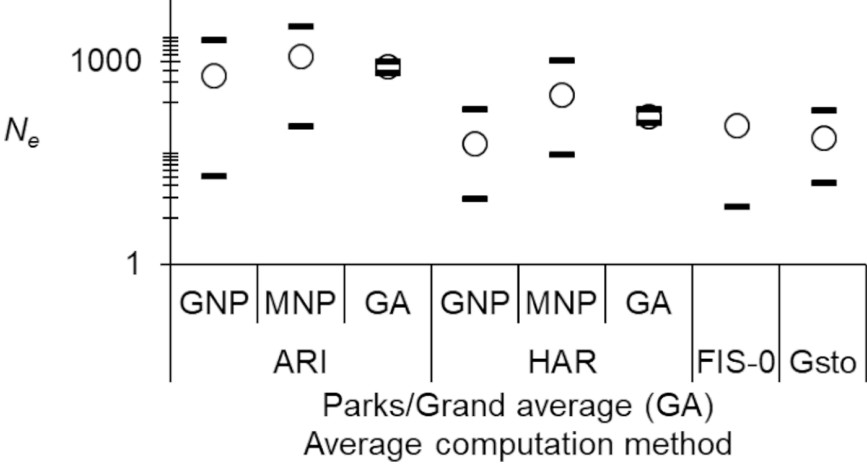

**Fig 4. Effective population sizes of *Glossina brevipalpis* from Gorongosa National Park (GNP) and Maputo National Park (MNP), averaged across methods and grand averages across parks (GA).** We present results obtained with arithmetic means (ARI) of $N_e$ (empty circles) and minimum and maximum values (black dashes) (minmax), with harmonic means (HAR) of $N_e$ (empty circles) and 95% confidence intervals (95%CI) and the result obtained from the intercept of the regression $F_{IS} \sim p_{nulls}$ (FIS-0, "infinite upper bound not shown) and its 95%CI for the present study. Arithmetic mean and minmax of values obtained in Gstöttenmayer et al., (2023) [31] (Gsto) are presented for comparison. The ordinates are in log scale.

## Bayesian clustering and neighbour-joining trees

Bayesian clustering expectedly produced an optimal partition with two clusters, but with multiple miss-assignments (Fig 5). Upon further inspection, these miss-assigned individuals seemed to contain several missing genotypes. Indeed, a correlation test between the probability of belonging to the park of origin and the number of missing data in each individual fly happened to be highly significant ($\rho = -0.1804$, $p$-value = 0.0002). We thus reran STRUCTURE keeping individuals with the fewest number of missing data. Exploring data with individuals with at least less than 40% missing genotypes still provided a significant signature ($\rho = -0.1808$, $p$-value = 0.0002). With 30% missing data, the correlation was not significant anymore ($\rho = -0.0633$, $p$-value = 0.136). We thus concluded that miss-assigned individuals with more than two missing data (i.e., more than 25% of the Multilocus genotype) rather corresponded to errors, while other misplaced individuals (with a 75% complete genotype at least), probably corresponded to individuals that inherited genes from recent immigrants or even represented recently introduced individuals (if with more than 90% genomic match). This was confirmed by the fact that even individuals with complete genotypes could be miss-assigned (S1 Fig). Individuals with no or a single missing genotype and with more than 90% assignment to the wrong park would correspond to recently introduced individuals (all in GNP). Other miss-assigned individuals would then correspond to individuals that inherited genes from more or less recent immigrants (14 in GNP and 6 in MNP). We also ran STRUCTURE with only one or no missing data allowed (S1 Fig). Miss-assigned individuals can still be met, though "recent" immigrants cannot be detected anymore with complete genotypes.

A Neighbour-joining approach on all individuals corrected for null alleles (S2 Fig) or on individuals with complete genotypes (S3 Fig), confirmed that some miss-assigned individuals were indeed genetically close to individuals from the other park.

## Discussion

During this study, we faced many genotyping problems, with a substantial number of missing genotypes at several loci. Most missing genotypes were due to PCR failures and not to null alleles, as Gb35 was the only locus for which missing data corresponded to null homozygotes. Fortunately, this did not alter most of our results as we could demonstrate that heterozygote deficits were fully explained by null alleles. We also could compute genetic differentiation accordingly. Nevertheless, during the Bayesian clustering process, such missing data significantly altered the probability of assignment of the most affected individuals to their population. Indeed, only the dataset with individuals with less than 25% of

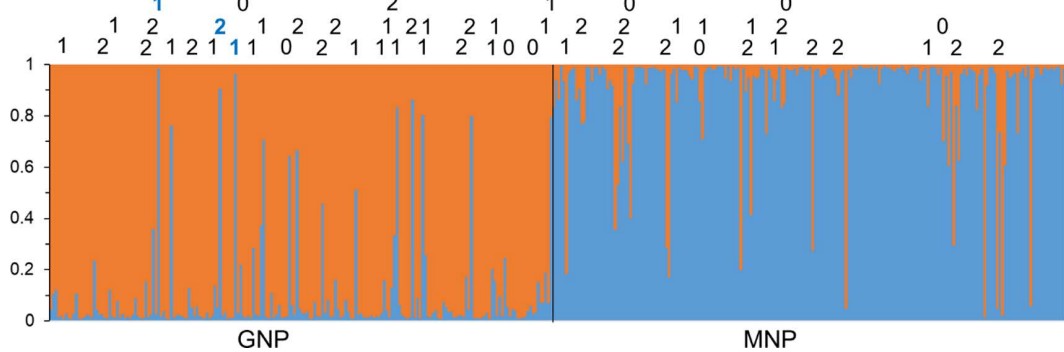

**Fig 5. Probabilities of assignment of *Glossina brevipalpis* individuals from the two parks of Mozambique, Gorongosa National Park (GNP – orange) and Maputo National Park (MNP – blue), after Bayesian clustering and for the optimal partition obtained (K =2).** Information on the top indicate the number of missing genotypes for the individuals that were the least assigned to their park of origin, with a threshold of 0.9, and displaying less than three missing genotypes. Individuals in bold blue are more than 90% miss-assigned and assimilated to new immigrants from MNP in GNP. Numbers in black are interpreted as introgressed with alien alleles inherited from some immigrant ancestors in the past.

missing genotypes (i.e., one or two missing loci) did not produce a significant effect of missing genotypes on probabilities of assignment. It means that the algorithm applied by this software used missing genotypes as informative to build the clusters.

Effective population sizes appeared substantial ($N_e$ = 844 in minmax = [676, 1012]), though probably underestimated. These appeared much larger than previously reported ($N_e$ = 76 in minmax = [16, 192]) [31], to what was predicted by the intercept of the regression $F_{IS} \sim p_{nulls}$ or if we use Waples' method. One hundred probably represents an extreme lower limit, while 1000 probably corresponds to the upper one.

Subdivision between parks was highly significant. With a two-subpopulation model, this subdivision appeared compatible with the number of immigrants $N_e m$ = 0.35 in 95% CI = [0.20, 0.68] exchanged per generation. Translated in years, this would mean 2 in 95% CI = [1, 4] individuals per year, exchanged between the two parks, or a total separation dating to 1936 on average with a time window between 1858 and 1966. This would be in line with both the rinderpest pandemic (1898–1901) and the last vector control campaign, which was implemented between 1948 and 1970 in the area [5,12–14,83]. Nevertheless, no signature of any bottleneck could be assessed. This does not confirm a drastic decrease in tsetse populations following any of these two events. This may be due to immigration events from other areas not included in our study, which may have brought enough exogenous alleles to wipe out any bottleneck signature. According to published data from the last 30 years [18], no *G. brevipalpis* subpopulations were reported that could link GNP and MNP in a step-by-step model. However, considering that very limited data are available from that region, this lack of reporting does not exclude the possibility that such pockets might exist and could be discovered in the future. Historical records identified a *G. brevipalpis* population pocket in Mutamba River valley near Inhambane city on the coast [84]. To this day, this area has favourable conditions for *G. brevipalpis* [15], but no recent reports indicate the presence of this species. A thorough survey of these historical sites needs to be conducted. Accordingly, Bayesian clustering did not support that total isolation between the two subpopulations occurred around 1946, nor that step-by-step dispersal could take place between undocumented subpopulations connecting the two parks. Indeed, the most probable interpretation of our results is the direct exchange of (very) rare immigrants, i.e., four per year on average at most. This was confirmed by using a Neighbour-joining approach, which confirmed that some miss-assigned individuals were indeed genetically close to individuals from the other park.

Effective population size and immigration could be translated into no more than 2 km distance dispersal per generation between the two parks, which appeared much smaller than the 24 km that these flies are able to travel within a park. The difference in dispersal distance within and between the parks illustrates how low survival is for tsetse flies travelling from GNP to MNP (or back). The maximum dispersal distance ever reported for a tsetse fly was 25 km in 24 days for *G. tachinoides*, after a mark-release-recapture study [85], and around 40 km/generation for *G. palpalis gambiensis* with population genetics data [86]. Consequently, it appears highly unlikely that immigrants could have crossed the 840 km separating the two parks on their own. The motorized anthropochory of these flies may be an alternative explanation for the flow of individuals between these two extremely distant populations.

We know that the recent introduction of wild animals occurred from one of these two parks to the other using trucks or airplanes that may have represented shelter, food and easy transport for *G. brevipalpis* [87,88]. For example, in 2019 waterbucks, warthogs, and oribi were transported to MNP from GNP and more recently, in 2021, a pack of African wild dogs were translocated from the buffer zone of MNP to GNP (M. Stalmans, personal communication, April 16, 2025). An informal inquiry at the National Directorate of Livestock Development within the Ministry of Agriculture, Environment and Fisheries provided evidence of livestock transportation (cattle and goats) between the two provinces (Maputo and Sofala) where the parks are located (J. Fabbri, personal communication, April 16, 2025). This can be related to long, though much smaller, range dispersal distances that were recently documented in other tsetse fly species [22,86,89]. Therefore, we advise considering that such a phenomenon may account for reinvasions over very long distances.

## Conclusion

The most relevant result obtained during this work is that, despite the existence of two distant and supposedly isolated populations, there is strong evidence indicating the exchange of rare individuals. This could be due to undiscovered pocket populations between the two parks, if so, a detailed entomological survey of possible habitats between the two parks needs to be conducted. A more probable explanation of our findings is that tsetse flies may have been moved between parks via motorized human transport means. Such passive dispersal should thus be investigated more thoroughly in future studies. We also recommend that a similar study be carried out on *G. austeni* populations in Mozambique that show a similar distribution pattern to that of *G. brevipalpis*. Such a study might further elucidate the tsetse dynamics between the centre and south of the country and assist in the development of efficient control strategies. If the eradication of the southern tsetse belt in MNP was attempted, the possibility of re-invasion of the tsetse via motorized human transport means needs being taken in to account as well as the exploration of other sites within a 30 km radius to validate that no sources of re-invasion exist around this park.

## Supporting information

**S1 Table. Trapping sites with GPS coordinates, the number of captured Glossina brevipalpis and number of genotyped tsetse at Gorongosa National Park and Maputo National Park in Mozambique.**
(XLSX)

**S2 Table. Dataset of tsetse genotypes.** Three hundred and ninety-six (396) *Glossina brevipalpis* genotypes collected at Gorongosa National Park (196) and Maputo National Parks (200) in Mozambique.
(TXT)

**S3 Table. The 10 polymorphic microsatellite loci used for genotyping Mozambican *Glossina brevipalpis* and associated genetic population parameters.** Sample size (genotypes per locus), number of alleles, allele size range, Write's fixation index in individuals as compared to subsamples ($F_{IS}$), expected heterozygosity ($H_{exp}$), Nei's unbiased expected heterozygosity ($H_S$), and observed heterozygosity ($H_O$). *Loci removed from final analyses.
(XLSX)

**S1 Fig. Structure results on *Glossina brevipalpis* from Gorongosa National Park (GNP) and Maputo National Park (MNP) in Mozambique for individuals with no more than one missing genotype out of eight (top) or without missing genotype (bottom).** Numbers on the top indicate the number of missing genotypes for individuals assigned to their park of origin with less than 50% probability. Average probabilities of assignment to the park of origin was 0.9234 and 0.7811 for One missing and No missing datasets, respectively.
(TIF)

**S2 Fig. A Neighbour-Joining tree on all individuals of *Glossina brevipalpis* from Gorongosa National Park and Maputo National Park in Mozambique.** The tree is based on Cavalli-Sforza and Edward's chord distance corrected for null alleles, built with MEGA X and edited with iTOL V6.
(PDF)

**S3 Fig. Neighbour-Joining trees of *Glossina brevipalpis* from Gorongosa National Park (orange) and Maputo National Park (blue) in Mozambique for individuals with complete genotypes.** These trees are based on Cavalli-Sforza and Edward's chord distance corrected for null alleles (left), or on Nei's $D_A$ genetic distance (right). These were built with MEGA X and PopTree, respectively, and edited in iTOL V6. Probabilities in black are the assignment of corresponding individuals to their park of origin.
(TIF)

## Acknowledgments

We would like to acknowledge the Biotechnology Centre of Eduardo Mondlane University and INTERTRYP IRD/CIRAD for using their laboratories and for their continued partnership throughout this study. We would also like to acknowledge Keila Zandamela and Edmilson Philimone for their assistance and dedication during this project. Authors would like to thank Marc Stalmans of Gorongosa National Park and José Fabbri of the National Directorate of Livestock Development (the Ministry of Agriculture, Environment and Fisheries, Mozambique) for providing information about the transport of livestock and wildlife within the country.

More details about the COMBAT project can be found in Boulangé et al. [90].

## Author contributions

**Conceptualization:** Denise R. A. Brito, Fernando C. Mulandane, Luis Neves.

**Data curation:** Denise R. A. Brito, Adeline Ségard, Sophie Ravel, Thierry De Meeûs.

**Formal analysis:** Denise R. A. Brito, Thierry De Meeûs.

**Funding acquisition:** Fernando C. Mulandane, Luis Neves.

**Investigation:** Denise R. A. Brito, Fernando C. Mulandane, Thierry De Meeûs.

**Methodology:** Denise R. A. Brito, Adeline Ségard, Fernando C. Mulandane, Nióbio V Cossa, Hermógenes N. Mucache, Sophie Ravel, Thierry De Meeûs.

**Project administration:** Fernando C. Mulandane.

**Resources:** Nióbio V Cossa, Hermógenes N. Mucache, Sophie Ravel.

**Software:** Denise R. A. Brito, Thierry De Meeûs.

**Supervision:** Adeline Ségard, Fernando C. Mulandane, Sophie Ravel, Thierry De Meeûs, Luis Neves.

**Validation:** Adeline Ségard, Fernando C. Mulandane, Sophie Ravel, Thierry De Meeûs, Luis Neves.

**Visualization:** Adeline Ségard, Sophie Ravel, Thierry De Meeûs, Luis Neves.

**Writing – original draft:** Denise R. A. Brito, Thierry De Meeûs.

**Writing – review & editing:** Denise R. A. Brito, Adeline Ségard, Sophie Ravel, Thierry De Meeûs, Luis Neves.

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
