## [Decision Letter · Decision Letter 0]

26 Mar 2025

Dear Dr. De Meeûs,

Thank you for submitting your manuscript to PLOS Neglected Tropical Diseases. After careful consideration, we feel that it has merit but does not fully meet PLOS Neglected Tropical Diseases's publication criteria as it currently stands. Therefore, we invite you to submit a revised version of the manuscript that addresses the points raised during the review process.

Please submit your revised manuscript within 30 days May 25 2025 11:59PM. If you will need more time than this to complete your revisions, please reply to this message or contact the journal office at plosntds@plos.org. Please include the following items when submitting your revised manuscript:

Response to Reviewers

* A marked-up copy of your manuscript that highlights changes made to the original version. You should upload this as a separate file labeled 'Revised Manuscript with Track Changes '.

* An unmarked version of your revised paper without tracked changes. You should upload this as a separate file labeled 'Manuscript '.

We look forward to receiving your revised manuscript.

Kind regards,

Shaden Kamhawi

co-Editor-in-Chief

Paul Brindley

co-Editor-in-Chief

**Journal Requirements:**

1) Please provide an Author Summary. This should appear in your manuscript between the Abstract (if applicable) and the Introduction, and should be 150-200 words long. The aim should be to make your findings accessible to a wide audience that includes both scientists and non-scientists. Sample summaries can be found on our website under Submission Guidelines:

3) Some material included in your submission may be copyrighted. According to PLOSu2019s copyright policy, authors who use figures or other material (e.g., graphics, clipart, maps) from another author or copyright holder must demonstrate or obtain permission to publish this material under the Creative Commons Attribution 4.0 International (CC BY 4.0) License used by PLOS journals. Please closely review the details of PLOSu2019s copyright requirements here: PLOS Licenses and Copyright. If you need to request permissions from a copyright holder, you may use PLOS's Copyright Content Permission form.

Potential Copyright Issues:

i) Figure 1. Please (a) provide a direct link to the base layer of the map (i.e., the country or region border shape) and ensure this is also included in the figure legend; and (b) provide a link to the terms of use / license information for the base layer image or shapefile. We cannot publish proprietary or copyrighted maps (e.g. Google Maps, Mapquest) and the terms of use for your map base layer must be compatible with our CC BY 4.0 license.

4) Please amend your detailed Financial Disclosure statement. This is published with the article. It must therefore be completed in full sentences and contain the exact wording you wish to be published.

**Comments to the Authors:**

**Please note that one of the reviews is uploaded as an attachment.**

**Reviewers' comments:**

Reviewer's Responses to Questions

**Key Review Criteria Required for Acceptance?**

**Methods**

-Are the objectives of the study clearly articulated with a clear testable hypothesis stated?

-Is the study design appropriate to address the stated objectives?

-Is the population clearly described and appropriate for the hypothesis being tested?

-Is the sample size sufficient to ensure adequate power to address the hypothesis being tested?

-Were correct statistical analysis used to support conclusions?

-Are there concerns about ethical or regulatory requirements being met?

Reviewer #1: The manuscript clearly outlines the study’s objectives, which focus on understanding genetic differentiation and gene flow in Glossina brevipalpis populations from two national parks in Mozambique. The hypothesis is testable, and the study design, involving genetic data collection and appropriate statistical analysis, is well-suited to address these objectives. The population is clearly described, with a focus on distinct populations from GNP and MNP, making it relevant to the hypothesis. The statistical methods, including Bayesian clustering, FST, and AMOVA, are appropriate and well-justified for investigating genetic structure.

Reviewer #2: It seems that only individuals with missing genotypes are miss-assigned in the STRUCTURE analysis. Therefore I would expect you to rerun all the analyses (not only STRUCTURE) with a dataset excluding individuals with high levels of missing genotypes. Did you do that ? What is the impact on the results ?

Just as suggestions (not formally required for publication):

Did you analyse your dataset taking into account the sex of flies to analyse sex-biased dispersal ?

Did you test isolation by distance (inside the two parks ?)

**Results**

-Does the analysis presented match the analysis plan?

-Are the results clearly and completely presented?

-Are the figures (Tables, Images) of sufficient quality for clarity?

Reviewer #1: The analysis pgenerally aligns with the analysis plan, though there are some instances where the handling of missing data and specific statistical choices (e.g., coding missing genotypes as null homozygotes) could be more clearly explained. The results are presented in a clear and detailed manner, with the necessary statistical values provided to support the conclusions. The figures, including tables and images, are of sufficient quality for clarity, but they may benefit from additional explanatory captions or in-text references (e.g. Figure 1).

Reviewer #2: Supplementary Table S1 is not easy to read. Could you please provide a summary table with the locations GPS points, number of traps, number of individuals trapped and genotyped (males and females) ?

In the results section, could you provide a table with the main statistics (Sample size, number of alleles, observed and expected heterozygosity, Fis) ? A figure with Fis for each locus would also help (in supplementary maybe).

**Conclusions**

-Are the conclusions supported by the data presented?

-Are the limitations of analysis clearly described?

-Do the authors discuss how these data can be helpful to advance our understanding of the topic under study?

-Is public health relevance addressed?

Reviewer #1: The conclusions are generally supported by the data presented, with the observed genetic differentiation between populations and evidence of rare immigration between parks being consistent with the findings. However, some of the conclusions, such as the impact of motorized transport on tsetse dispersal, would benefit from further clarification and supporting evidence.

The limitations of the analysis are acknowledged, particularly in relation to the challenges posed by missing genotypes and the potential bias in the Bayesian clustering process.

Reviewer #2: In the discussion section, I expected that you compare your results with previously published ones on tsetse flies. Are they comparable ? Different ? Why ?

It is not clear to me what recommendations emerge from your study. I guess that the main message is that there is a risk of « reinvasions over very long distance ». Could you expand a bit on that ?

**Editorial and Data Presentation Modifications?**

Reviewer #1: Minor revision

Reviewer #2: (No Response)

**Summary and General Comments**

Reviewer #1: The paper presents an important study on the genetic differentiation of Glossina brevipalpis populations in Mozambique using microsatellite markers. Understanding gene flow and population structure is essential for effective vector control strategies. The paper is very solid though the findings are entirely expected given the spatial scale of the sampling and the existence of obvious variance (deduced from the authors own admissions). One thing that I have to give it up to the authors is the degree of thoroughness and robustness in the statistical approaches; the analyses were well handled. I have few comments and/or specific questions.

1. The abstract can and should conclude with a stronger statement on the implications for vector control (if any).

2. Line 28: The last sentence of the abstract offers a great implication for further research, but might benefit from slight rewording for clarity: “We suggest that undocumented means of dispersal, such as translocation through human-driven animal movement, should be explored in future studies investigating Glossina populations”

3. The human-driven transport hypothesis is plausible but not directly tested. The authors should either provide supporting evidence (e.g., known livestock transport routes) or acknowledge this as a limitation.

4. Line 100-101: What was the average distance between the traps in each location?

5. Make sure that each figure has a detailed caption that helps the reader understand what is being presented. For example, in the Figure 1; state clearly what the G1-G4 and M1-M4) represent; it’s not very obvious that it relates to the Park names!

6. Line 110. 402 samples were selected for genotyping based on the location and condition of specimen. What happened to the rest of the samples??

7. Line 180: Is it “recoded” or “recorded”? This has been repeated severally if it’s not “recoding”

8. Line 96. According to the authors’ own descriptions, it appears several species of Glossina occur sympatrically in the sampled areas. How did you definitively identify brevipalpis for genotyping? This is not described in the methods.

9. Line 96-98: Just out of curiosity, why did you use two different trap types (NGU and H-traps) for sampling?

10. Lines 343-346: How might these mis-assignments affect the inferences about migration and gene flow?

11. Lines 390-392. Again on the suggestion of motorized human transport as being responsible for the exchange of individuals between the parks, this is an interesting hypothesis that warrants further exploration. Could the authors provide a more detailed discussion of how such transport could occur and whether there are documented instances of tsetse flies being accidentally transported by humans, vehicles, or animals?

Reviewer #2: This manuscript deals with the population genetics of a Glossina brevipalpis, a vector of Trypanosoma parasites, in Mozambique. The authors sampled and genotyped about 400 tsetse flies in two National Parks in Mozambique with 10 microsatellite markers. Tsetse distribution in Mozambique is discountinued with a southern fly belt (where Maputo National Park is) and a central belt (where Gorongosa National Park is). The main objective of this study is therefore to understand gene flow between the two parks.

The authors acknowledge that they « faced many genotyping problems with a substantial number of missing genotypes at several loci ». Nevertheless, they draw convincing conclusion on the high genetic differentiation between the two parks, with low migration rates (2 individuals per year). They discuss the hypothesis of pocket populations between the two parks or human-driven transportation of tsetse flies.

I think that the results section is a bit hard to follow, very technical while the introduction and/or discussion miss a bit of comparisons with what has already been published on tsetse flies population genetics.

Overall, I believe that this work brings useful information on the dispersal of tsetse flies, which could help designing more efficient control strategies.

My main concern is about the quality of the genotype data. Quality testing is presented after analyses in the Mat & Meth and mixed with analyses in the Results. It both complicates reading and bring doubts on the obtained results. I believe that the authors did the right tests and analyses but the presentation is not so clear. Also this paper lacks a complete discussion about published results on tsetse flies population genetics (to compare with the present results) and about the consequences of the results on vector control.

PLOS authors have the option to publish the peer review history of their article (what does this mean? ). If published, this will include your full peer review and any attached files.

**Do you want your identity to be public for this peer review?** For information about this choice, including consent withdrawal, please see our Privacy Policy .

Reviewer #1: **Yes: ** Robert Opiro

Reviewer #2: **Yes: ** Hélène Jourdan-Pineau

**Figure resubmission:**
---

## [Editor Report · Decision Letter 1]

12 May 2025

Dear Dr De Meeûs,

We are pleased to inform you that your manuscript 'Unravelling genetic differentiation between Glossina brevipalpis populations from two distant National Parks in Mozambique' has been provisionally accepted for publication in PLOS Neglected Tropical Diseases.

Best regards,

Brian L Weiss

Academic Editor

Amy Morrison

Section Editor

Shaden Kamhawi

co-Editor-in-Chief

Paul Brindley

co-Editor-in-Chief

---

## [Editor Report · Acceptance letter]

Dear Dr De Meeûs,

We are delighted to inform you that your manuscript, "Unravelling genetic differentiation between Glossina brevipalpis populations from two distant National Parks in Mozambique," has been formally accepted for publication in PLOS Neglected Tropical Diseases.

Best regards,

Shaden Kamhawi

co-Editor-in-Chief

Paul Brindley

co-Editor-in-Chief
